# Date Palm Pollen Extract Avert Doxorubicin-Induced Cardiomyopathy Fibrosis and Associated Oxidative/Nitrosative Stress, Inflammatory Cascade, and Apoptosis-Targeting Bax/Bcl-2 and Caspase-3 Signaling Pathways

**DOI:** 10.3390/ani11030886

**Published:** 2021-03-20

**Authors:** Samar S. Elblehi, Yasser S. El-Sayed, Mohamed Mohamed Soliman, Mustafa Shukry

**Affiliations:** 1Department of Pathology, Faculty of Veterinary Medicine, Alexandria University, Alexandria, Edfina 22758, Egypt; 2Department of Forensic Medicine and Toxicology, Faculty of Veterinary Medicine, Damanhour University, Damanhour 22511, Egypt; elsayed-ys@vetmed.dmu.edu.eg; 3Clinical Laboratory Sciences Department, Turabah University College, Taif University, P.O. Box 11099, Taif 21944, Saudi Arabia; mmsoliman@tu.edu.sa; 4Department of Physiology, Faculty of Veterinary Medicine, Kafrelsheikh University, Kafrelsheikh 33516, Egypt

**Keywords:** cardiac injury markers, oxidative stress, histopathology, Bcl-2, Bax, TGF-β1, date palm (pollen extract), doxorubicin

## Abstract

**Simple Summary:**

The use of date palm pollen ethanolic extract (DPPE) is a conventional approach in improving the side-effects induced by Doxorubicin (DOX).DPPE mitigated DOX-induced body and heart weight changes and ameliorated DOX-induced elevated cardiac injury markers. In addition, serum cardiac troponin I concentrations (cTnI), troponin T (cTnT), and N-terminal NBP and cytosolic (Ca^+2^) were amplified by alleviating the inflammatory and oxidative injury markers and decreasing histopathological lesions severity. DPPE decreased DOX-induced heart injuries by mitigating inflammation, fibrosis, and apoptosis through its antioxidant effect. To reduce DOX-induced oxidative stress injuries and other detrimental effects, a combined treatment of DPPE is advocated.

**Abstract:**

Doxorubicin (DOX) has a potent antineoplastic efficacy and is considered a cornerstone of chemotherapy. However, it causes several dose-dependent cardiotoxic results, which has substantially restricted its clinical application. This study was intended to explore the potential ameliorative effect of date palm pollen ethanolic extract (DPPE) against DOX-induced cardiotoxicity and the mechanisms underlying it. Forty male Wistar albino rats were equally allocated into Control (CTR), DPPE (500 mg/kg bw for 4 weeks), DOX (2.5 mg/kg bw, intraperitoneally six times over 2 weeks), and DPPE + DOX-treated groups. Pre-coadministration of DPPE with DOX partially ameliorated DOX-induced cardiotoxicity as DPPE improved DOX-induced body and heart weight changes and mitigated the elevated cardiac injury markers activities of serum aminotransferases, lactate dehydrogenase, creatine kinase, and creatine kinase-cardiac type isoenzyme. Additionally, the concentration of serum cardiac troponin I (cTnI), troponin T (cTnT), N-terminal pro-brain natriuretic peptide (NT-pro BNP), and cytosolic calcium (Ca^+2^) were amplified. DPPE also alleviated nitrosative status (nitric oxide) in DOX-treated animals, lipid peroxidation and antioxidant molecules as glutathione content, and glutathione peroxidase, catalase, and superoxide dismutase activities and inflammatory markers levels; NF-κB p65, TNF-α, IL-1β, and IL-6. As well, it ameliorated the severity of histopathological lesions, histomorphometric alteration and improved the immune-staining of the pro-fibrotic (TGF-β1), pro-apoptotic (caspase-3 and Bax), and anti-apoptotic (Bcl-2) proteins in cardiac tissues. Collectively, pre-coadministration of DPPE partially mitigated DOX-induced cardiac injuries via its antioxidant, anti-inflammatory, anti-fibrotic, and anti-apoptotic potential.

## 1. Introduction

Doxorubicin (Adriamycin^®^), an anthracycline chemotherapeutic medication, has been effective against several types of malignancies since the 1960s [1,2]. It is the most valuable cytotoxic medication approved by oncologists in tandem with other anti-tumor medications or radiation and surgery [3]. It is highly potent and effective against solid tumors, i.e., breast, lung, bladder, gastrointestinal, thyroid, testicular, and ovarian carcinoma [4,5]. It is also used for treating hematological cancers, i.e., Hodgkin’s and non-Hodgkin’s lymphoma and pediatric leukemia [2,3]. Two proposed mechanisms for DOX antineoplastic effects have been reported [6]. The first one is through DNA chelation as DOX interacts with DNA, inhibits topoisomerase-II progression, and hinders DNA repair, which triggers DNA damage and cell death [7]. The second mechanism includes reactive oxygen species creation (ROS) and oxidative stress induction [8]. In vivo, DOX is metabolized into an unstable semiquinone, which is transformed back to DOX in a reaction that discharges ROS and reactive nitrogen species (RNS), causing lipid peroxidation, cell membrane, DNA, and proteins damages, and prompts apoptotic pathways of cell downfall to kill cancer cells [8,9]. Genes that can regulate this pathway include those involved with the oxidation outcome (xanthine oxidase, NADH dehydrogenases, and nitric oxide synthases) and those which disable free radicals, involving catalase (CAT), superoxide dismutase (SOD) and glutathione peroxidase (GPx) [9,10]. Nevertheless, such effects are not selective for cancer cells alone as the same mechanisms can also affect healthy cells [11,12].

It has been reported that DOX administration can induce structural and functional cardiac alterations, i.e., ventricular distension, diminished output, systolic and diastolic disturbance [13,14], congestive heart failure (CHF), left ventricular remodeling, and cardiomyopathy [15,16]. The detailed mechanisms behind DOX-induced cardiac injury have not been elucidated, but it is possibly involved with several paths. Previous studies reported that DOX-induced cardiotoxicity involved the production of oxidative ROS [17]. As DOX enters the body, it binds tightly to cardiolipin present in the inner mitochondrial sheath [18], accumulates in mitochondria, and affects the electron transport chain creation of ROS and RNS [19]. Later, they aggravate mitochondrial and cellular membrane damage and diminish the antioxidant defense system [20,21], subsequently leading to cell apoptosis [21].

Mitochondrial damage can also initiate intracellular Ca^2+^ imbalance [22], which further affects the apoptosis paths and causes myocardial cell death [23]. DOX also interferes with iron regulation [24], up-regulates NF-κB expression, which consequently causes the release of pro-inflammatory cytokines, i.e., tumor necrosis factor-alpha (TNF-α), interleukin-1 beta (IL-1β), and interleukin-6 (IL-6), and triggers vascular and cardiac inflammatory reaction [25] and exaggerates their downstream apoptotic pathways [26]. In addition, oxidative stress activates several pro-fibrogenic factors, which enhances the accumulation of extracellular matrix and development of cardiac fibrosis [27], remodeling [28], and eventual cardiac dysfunction [29]. DOX is still in use. To remain an efficient anticancer medication, it is essential to find appropriate new therapeutic agents to serve as adjuvants to mitigate DOX-induced cardiotoxicity. Several therapeutic strategies were developed to minimize DOX-induced oxidative injury, inflammation, DNA damage, and apoptosis. However, most of them interfere with DOX’s therapeutic effects, limiting their clinical use for cardio-protection against DOX-induced cardiotoxicity [30] and up till now no specific effective and safe drugs have been found.

Phytomedicine is one of the strategies that focus on chemical substances naturally present in plants to improve health conditions and prevent, manage, and treat many diseases. Plant phenolics are natural antioxidant agents that embrace an electron that forms comparatively stable phenoxyl radicals and, consequently, interrupt the redox reactions within the cells [31]. They were also found to activate a cellular redox defense mechanism by stimulating endogenous antioxidant fractions [32] and keep the cells from xenobiotic oxidative stress, DNA impairment, and apoptosis [33]. Much attention has been given to using plant chemicals as a defensive strategy to resolve cardiotoxicity triggered by DOX [34,35,36].

Date palm pollen (DPP) is a powder formed from date palm (*Phoenix dactylifera* L.) male reproductive cells. It has been utilized by the initial Chinese and the primeval Egyptians as a regenerating factor and worldwide as a dietary supplement [37,38]. Yearly, approximately one thousand tons of DPP are created in Arabic areas [39]. DPP is rich in many health-promoting factors, i.e., flavonoids and volatile unsaturated fatty acids [40,41]; that have strong antioxidant properties in scavenging free radicals [41,42]. In addition, DPP has anti-inflammatory, anti-coccidial, aphrodisiac, anti-apoptotic actions, and is a hepatoprotective agent [43,44,45,46]. Egyptian DPP has been proven to have a vast range of nutritionally and biochemically bioactive constituents, i.e., essential and non-essential amino acids, nucleic acids, different carbohydrates, trace elements, minerals, and vitamins. It also contains important phenolic compounds, including gallic, caffeic, coumaric, cinnamic, ferulic acids, catechin, rutin, quercetin, and naringenin propyl gallate. It also contains saturated (arachidic, capric, lauric, myristic, palmitic, and stearic) and unsaturated (arachidonic, linoleic, linolenic, oleic and palmitoleic) fatty acids, ω3, ω6 [47] and a lot of enzymes and cofactors [38,48]. Furthermore, Egyptian DPP has estrogenic substances, i.e., estriol, estradiol (E2), and estrone, which were recognized to alleviate male subfertility problems through their gonadotrophic activity [49].** This study was intended to explore the potential ameliorative effect of date palm pollen ethanolic extract (DPPE) against DOX-induced cardiotoxicity and the mechanisms underlying it.

## 2. Materials and Methods

### 2.1. Chemicals, Kits, and Reagents

Doxorubicin HCl (Adricin^®^) injectable solution was procured from EIMC United Pharmaceuticals (Badr City, Cairo, Egypt). Commercially available kits for the measurement of ALT, AST, LDH, CK, CK-MB, GSH, GPx, CAT and SOD pursuits, and NO and MDA contents were obtained from Biodiagnostic Co. (Cairo, Egypt). Rat specific ELISA kits for cTnI and cTnT levels were gained from MyBiosource, Inc. (San Diego, CA, USA). ELISA kit for NT-proBNP was obtained from CUSABIO (Hubei, China). A commercially available colorimetric kit for Ca^+2^ was purchased from Elabscience Co. (Houston, TX, USA). Rats-specific ELISA kits for IL-6, IL-1β, and TNF-α were bought from BD Biosciences (San Jose, CA, USA). Masson’s trichrome was purchased from (Sigma Aldrich, St. Louis, MI, USA). NF-κB p65 total ELISA Kit, hydroxyproline colorimetric assay kit, hematoxylin and eosin stain (H&E), rabbit polyclonal anti-TGF-β1 antibody (Product# ab25121), rabbit polyclonal anti-cleaved caspase-3 antibody (Product# ab4051) and rabbit monoclonal anti-Bax antibody E63 (Product# ab32503) were purchased from Abcam Co. (Cambridge Science Park, Cambridge, UK). Rabbit polyclonal anti-Bcl-2 antibody (Product# PA5-27094) was obtained from Thermo Fisher Scientific Co. (Waltham, MA, USA).

### 2.2. Date Palm Pollen Grains Collection and Ethanolic Extract Preparation

Date palm pollen grains were gathered from *Phoenix dactylifera* L. in March 2020 from El-Beheira, Egypt. They are authenticated at the Department of Botany, Faculty of Science, Alexandria University. After collection, the pollen grains were dissected from the bark and washed with water, dried with air, and ground at room temperature using a grinder to fine powder kept at 4 °C until use.

Two hundred grams of DPP powder was extracted twice with 1600 mL of 80% ethanol for 24 h at room temperature. The extract was filtered in a Buchner funnel and then centrifuged at 5000 radius centrifugation force (RCF) for 30 min. The obtained supernatant was evaporated at 40 °C in a rotary evaporator under vacuum till complete dryness; then, the final dry extract and stock solution was preserved in dark glass bottles in the refrigerator at 4 °C for further analysis. The DPPE was re-dispersed in distilled H_2_O and orally intubated to treated rats using an intragastric tube at the time of experimentation.

### 2.3. Acute Oral Toxicity of DPPE “Median Lethal Dose, LD_50_”

Acute toxicity trial was carried out following the guidelines of the Organization for Economic Co-operation and Development [50] to evaluate the acute oral hazard of DPPE. The ‘Limit Test’ in the up and down procedure (UDP) was conducted to reduce the overall animals’ suffering. A maximum of 5 male Wister albino rats per group was administered sequentially with DPPE up to a test dose of 5000 mg/kg bw. Twenty-five adult male Wister albino rats (180 ± 10 g bw) were allocated randomly into 5 groups (5 rats each) and were acclimatized for 7 days. The rat groups were fasted overnight and the next morning, freshly prepared DPPE was orally administrated to groups 1–5 at doses of 1, 2, 3, 4, and 5 g/kg bw, respectively. The limit test involved dosing an animal with up to 5 g/kg bw. If the animal managed to survive, two extra animals were dosed. If both animals stayed alive, the LD_50_ was supposed to be higher than the limit dose, and the test was finished. The rats were observed every 2 h for 24 h, and again at 48 and 72 h to record any behavioral changes, signs of toxicity, and mortalities. The survived animals were observed for any delayed toxic signs or death for the next 14 days.

### 2.4. Animals Experimentation and Sampling

Forty adult male Wister albino rats weighing approximately 190 ± 10 g (10 weeks-old) were purchased from the Medical Research Institute, Alexandria University, Egypt. The rats were kept in stainless-steel boxes at controlled environment “temperature 25 ± 5 °C and humidity 55 ± 5%” with a 12 h light/dark cycle and free access to standard rat feed (El-Nasr Co., Cairo, Egypt) and water for 2 weeks before the experiment to follow-up normal growth and behavior. The animals were given humane treatment in compliance with the Institutional and National Procedures for the Care and Use of Experimental Animals (NIH). They were declared by the Local Committee of the Faculty of Veterinary Medicine, Alexandria University (Ethical Committee Approval Number: 2020/013/59) and ethical approval of Taif University (42-0081).

After acclimatization, the rats were randomly distributed into 4 equal groups (10 rats each). Group I (CTR) rats weighing approximately (192 ± 3.6) were orally intubated with 1 mL distilled water using a stomach tube daily for about 4 weeks. Group II (DPPE-treated) rats weighing approximately (190 ± 8.1) were orally intubated with DPPE at a dose of 0.5 g/kg bw daily for 4 weeks. Groups I and II were also intraperitoneally injected with 0.5 mL isotonic saline solution six times over the last two weeks of the experiment. Group III (DOX-treated) rats weighing approximately (195 ± 5.2) were orally intubated with 1 mL distilled water for 4 weeks and were DOX injected intraperitoneally at a dose of 2.5 mg/kg bw six times over the last two weeks of the experiment [51] with an accumulative dose of 15 mg/kg bw. Group IV (DPPE + DOX) weighing approximately (194 ± 4.5) rats obtained DPPE and DOX at the same dosage used in groups II and III. DPPE was administrated to rats an hour before DOX administration (Figure 1).

At the end of the experimentation, the animals were only allowed free access to water and fasted for 12 h. After that, they were weighed, and blood samples were obtained just before euthanasia from the retro-orbital plexus of the inner eye canthus under diethyl ether anesthesia. The collected blood was centrifuged for 10 min at 3000 rpm, and then the resulting sera samples were kept at −20 °C for further analysis. Subsequently, rats were euthanized by cervical dislocation. The heart was rapidly harvested, rinsed with saline, dried, weighted, and dissected. The cardiac specimens were immediately frozen and kept at −80 °C. In an ice-cold phosphate buffer saline, the frozen samples were thawed and homogenized. (0.1 M pH 7.4) utilizing a homogenizer with a Teflon pestle and then centrifuged at 5000× *g* for 15 min. at 4 °C. Aliquots of the supernatant were frozen at −80 °C for the chemical analysis. In neutral buffered formalin 10% solution, other heart specimens were immediately fixed for the histopathological and immunohistochemical assessment.

### 2.5. Assessment of the Body, Heart, and Relative Heart Weights

At the end of the experimentation, in each rat, the body and heart weight were recorded. The relative heart weights (RHW) were estimated using the following formula: $$\mathrm{RHW}=\frac{\mathrm{Heart}\;\mathrm{weight}\;\left(g\right)}{\mathrm{Bodyweight}\;\left(g\right)}\times 100$$

### 2.6. Assessment of Cardiac Injury Biomarkers and Cytosolic Calcium (Ca^+2^)

The serum ALT, AST, LDH, CK, and CK-MB levels were estimated as instructed by the manufacturers. The serum cTnI, cTnT, and NT-proBNP (a marker of heart failure) concentrations were also measured using the corresponding enzyme-linked immunosorbent assay (ELISA) kits using ELISA Plate Reader (Bio-Rad, Hercules, CA, USA). The supernatant obtained by centrifugation of cardiac tissue homogenate was used to evaluate the concentration of cytosolic ${\mathrm{Ca}}^{2+}$ using the ${\mathrm{Ca}}^{2+}$ colorimetric assay kit, as instructed by the manufacturer.

### 2.7. Estimation of Cardiac Nitro-Oxidative Stress and Lipid Peroxidation

The concentration of nitric oxide (NO) was assessed in the supernatants of the cardiac homogenates based on the enzymatic reduction of nitrate to nitrite. For nitrite detection, the colored azo dye product “Griess reaction” was spectrophotometrically monitored at 550 nm absorbance [52]. The levels of MDA [53] and GSH [54] and the activities of GPx [55], CAT [56], and SOD [57] were spectrophotometrically estimated in the cardiac tissue homogenates. The total protein content was also assessed [58].

### 2.8. Assessment of Inflammatory Markers

The cardiac total NF-κB p65, TNF-α, IL-1β, and IL-6 were measured in the supernatant obtained by centrifugation of cardiac tissue homogenate using the corresponding rat-specific ELISA kit following the manufacturer’s protocols.

### 2.9. Estimation of Hydroxyproline Content

Briefly, about 100 mg of the right ventricle was homogenized in double-distilled water. In a tightened screw-capped polypropylene vial, the tissue homogenates were mixed with conc NaOH (10 N) and then boiled for an hour at 120 °C. The alkaline lysate was ice-cooled, neutralized to pH 7.0, and centrifuged to get off supernatants. The hydrolysates were hot air-dried, chloramine T-oxidized, and finally reacted with Ehrlich’s reagent. The resultant colored product was measured at 560 nm absorbance, and the amount of hydroxyproline content was detected by comparing it with a standard curve [59].

### 2.10. Histopathological Assessment and Semi-Quantitative Scoring Approach

Cardiac samples were immediately fixed in phosphate-buffered formalin (10%, pH 7.4) after necropsy for 24 h, then were handled using the conventional paraffin embedding method. The 5 μm thick pieces were cut and placed on slides, deparaffinated in xylene, and rehydrated using decreasing concentrations of ethanol. One set of slides was hematoxylin and eosin (H&E)-stained for the routine histopathological setting. An additional set was Masson’s trichrome-stained for detecting the amount and distribution of collagen fibers [60]. Stained sections were blindly examined using light microscopes and photographed using a digital camera at a magnification of 400× (Nikon Corporation Co., Ltd., Tokyo, Japan).

To convey the occurrence and severity of the histopathological lesions, a semi-quantitative scoring approach was used. In each animal group, seven H&E-stained slides (one slide/rat) were examined, and 10 random fields per slide were used for grading the various pathological lesions in a blinded fashion. The severity of pathological lesions was asessed according to the percentage of tissue affected in the entire section as None (−): normal histology with zero immersion of the inspected field, Mild (+): 5–25% of the tested field was involved, Moderate (++): 26–50% of the inspected field was involved, Severe (+++): >50% of the examined field was applied. The incidence represented the number of lesion rats per total examined.

### 2.11. Immunohistochemical Assessment

According to Hsu, et al. [61], the immunodetection was assessed using four overlapping paraffin-embedded cardiac tissue sections. Sections were sliced at 4 µm thicknesses utilizing a microtome and put-on slides that are positively charged. Then, the sections were deparaffinized, rehydrated in xylene, then in different graded ethanol solutions and underwent antigen repossession using sodium citrate buffer (10 mM, pH 6.0) in the microwave at 105 °C for 10 min. Then, the activity of endogenous peroxidase was inhibited with $3{\%\;H}_{2}O_{2}$ for 10 min.; the non-specific proteins were blocked with 5% goat serum for 30 min at room temperature. The cardiac tissue slices were washed thrice in Dako Tris-buffered saline (TBS) and then incubated with the specific rabbit primary antibodies: polyclonal anti-TGF-β1 (dilution 1/200), anti-cleaved caspase-3 (dilution1/100), monoclonal anti-Bax (dilution1/250), and anti-Bcl-2 (dilution 1/100) at 4 °C overnight. In the negative control sections, normal IgG was substituted for the primary antibodies at the same concentration and antibody species. Following PBS washing, the tissue sections were incubated for an hour with goat anti-rabbit biotin-labeled secondary antibody, rinsed in PBS for 2 min. Then the sections were incubated with streptavidin-horseradish peroxidase reagent (VECTASTAIN1 Elite ABC kit, Vector Laboratories, Inc., Burlingame, CA, USA) at 37 °C for 20 min then washed with rinsing buffer and incubated with 3,3-diaminobenzidine tetrahydrochloride (DAB Substrate Kit, Thermo Fischer Scientific, Rockford, IL, USA) as the chromogen to developed peroxidase reaction. The sections were finally objected to Mayer’s Hematoxylin to augment the nuclear staining and mounted with di-poly cysteine xylene (DPX). All slides were assessed blindly and photographed using a digital camera.

### 2.12. Histomorphometric Assessment

The assessment was performed using H&E, Masson’s trichrome, and immunostained cardiac sections (one section from each rat and seven per group). The digital images (ten different fields per section at ×400 magnification power) were blindly analyzed using image analysis software (ImageJ Version 1.47, National Institutes of Health, Bethesda, MD, USA, wayne@codon.nih.gov. The ten values were averaged in each animal, and the average was used as individual sampling data.

Using H&E-stained sections, the cross-sectional area of cardiomyocytes was assessed in the left ventricular wall. Fifteen cardiomyocytes with a visible nucleus and intact cellular membrane were selected per field for the measurement and analysis [62].

Using Masson’s trichrome-stained sections [63], the collagen volume fraction (CVF %) and perivascular collagen area (PVCA %) percentage were estimated as per the following formulas: $$\mathrm{CVF}\;\left(\%\right)=\frac{\mathrm{Collagen}\;\mathrm{area}}{\mathrm{Total}\;\mathrm{area}}\times 100;$$
PVCA (%)=area occupied by the collagen/total area of the vessel section×100 .

Using images of immunostained slides, the area percentage (%) of TGF-β1, cleaved caspase-3, Bax, and Bcl-2 immunopositive cardiomyocytes were estimated as area percent (%) across ten different fields/sections [64].

### 2.13. Data Analysis

One-way Variance Analysis evaluated the numerical data estimation (ANOVA)test using SPSS data analysis software (Version 21; SPSS Inc., Chicago, IL, USA) and summarized it as means ± (SEM). Tukey’s post-hoc test was used to ascertain the statistical difference between experimental groups. * *p* < 0.05 was set as statistically significant.

## 3. Results

### 3.1. Median Lethal Dose, Mortality, and Survival Rates

Rats of DPPE groups did not exhibit any behavioral changes, toxic side-effects, or even mortalities after 24 h and 14 days post-treatment. Thus, dosing was ceased at 5 g/kg bw. Consequently, the LD_50_ of DPPE was evaluated to be more than 5 g/kg bw. The CTR and DPPE group did not exhibit any mortality all over the experimental period. However, the DOX group showed a scruffy appearance and exhibited 30% mortality (three dead rats), and DPPE + DOX group exhibited 10% mortality (one dead rat) (the data not shown).

### 3.2. Body, Heart, and Relative Heart Weights and Myocyte Cross-Sectional Area

As demonstrated in Table 1, CTR and DPPE groups exhibited a statistically non-significant change in the body weight, heart weight, RHW, and myocyte cross-sectional area. Conversely, the DOX group displayed a significant reduction in body weight (≈0.86-fold) and a substantial rise in the heart weight, RHW, and the myocyte cross-sectional area (1.6, 1.7, and 1.52-fold, respectively) compared to CTR values. Nevertheless, DPPE + DOX group exhibited a non-substantial rise in body weight (≈1.08-fold) and a considerable reduction in heart weight (≈0.76-fold), RHW (≈0.71-fold), and the myocyte cross-sectional area (≈0.73-fold) compared to DOX group values. In comparison with the CTR group values, DPPE + DOX group expressed a non-significant reduction in body weight (≈0.93-fold), a significant rise in the heart weight (≈1.22-fold), a non-significant increase in the RHW (≈1.2-fold), and a substantial increasing in the cardiomyocyte cross-sectional area (≈1.22-fold).

### 3.3. Cardiac Injury Biomarkers and Cardiac Cytosolic Calcium ($Ca^{2+}$)

As demonstrated in Table 2, the CTR and DPPE groups disclosed a statistically non-significant change in the activities of ALT, AST, LDH, CK, CK-MP, and the levels of cTnI, cTnT, NT-pro BNP, and cardiac ${\mathrm{Ca}}^{2+}$. Meanwhile, the DOX group exhibited a significant increase in ALT (≈1.52-fold), AST (≈1.42-fold), LDH (≈2.49-fold), CK (≈2.8-fold) and CK-MP (≈2.03-fold) activities, and cTnI (≈5-fold), cTnT (≈3.44-fold), NT-pro BNP (≈2.38-fold), and cardiac ${\mathrm{Ca}}^{2+}$(≈1.74-fold) levels as compared to CTR values. In contrast, DPPE + DOX group demonstrated a significant reduction in ALT (≈0.79-fold), AST (≈0.85-fold) LDH (≈0.73-fold), CK (≈0.7-fold) and CK-MP (≈0.72-fold) activities, and cTnI (≈0.57-fold) fold, cTnT (≈0.54-fold), NT-pro BNP (≈0.63-fold) and cardiac ${\mathrm{Ca}}^{2+}$ (≈0.82-fold) levels, as compared to DOX values. In addition, this group showed a significant increase in the activities of ALT (≈1.2-fold), AST (≈1.22-fold), LDH (≈1.83-fold), CK (≈1.97-fold), and CK-MP (≈1.48-fold), and the levels of cTnI (≈2.89-fold), cTnT (≈1.84-fold) NT-pro BNP and cardiac ${\mathrm{Ca}}^{2+}$(≈1.43-fold), as compared to the CTR values.

### 3.4. Cardiac Nitro-Oxidative Stress and Lipid Peroxidation

As demonstrated in Table 3, the CTR and DPPE groups revealed a non-significant change in the concentrations of NO, lipid peroxidation marker (MDA), and antioxidant parameters (GSH level and GPx, CAT, and SOD activities). Meanwhile, the DOX group disclosed a statistically substantial increase in the quantities of NO (≈3.61-fold), MDA (≈1.86-fold), and a significant reduction of the GSH level, and GPx, CAT, and SOD pursuits (≈0.46, 0.31, 0.54 and 0.31-fold, respectively) compared to the CTR values. Quite the opposite, the DPPE + DOX group demonstrated a considerable drop in the levels of NO (≈0.58-fold) and MDA (≈0.69-fold) and a significant rise in the GSH level and GPX, CAT, and SOD activities (≈1.67, ≈2.18, ≈1.41 and ≈2.28-fold, respectively) equated to the DOX group values. Compared to the CTR group values, DPPE + DOX group showed a significant rise in the levels of NO (≈2.11–fold), MDA (≈1.29–fold), and a considerable reduction in the GSH level, and GPx, CAT, and SOD pursuits (≈ 0.77, 0.68, 0.77 and 0.72-fold, respectively).

### 3.5. Inflammatory Markers

As shown in Table 4, the CTR and DPPE groups displayed a statistically non-significant change in the NF-κB p65, TNF-α, IL-1β, and IL-6 concentrations. Meanwhile, the DOX group exhibited a significant rise in the quantities of NF-κB p65 (≈2.63-fold), TNF-α (≈2.35-fold), IL-1β (≈1.74-fold), and IL-6 (≈1.85-fold) equated to CTR values. Conversely, DPPE + DOX group disclosed a significant reduction in the concentrations of NF-κB p65 (≈0.72-fold), TNF-α (≈0.63-fold), IL-1β (≈0.82-fold), and IL-6 (≈0.74-fold) compared to the DOX group values. Compared with CTR values, DPPE + DOX group exhibited a significant rise in the NF-κB p65 (≈1.9-fold), TNF-α (≈1.48-fold), IL-11β (≈1.43-fold), and IL-6 (≈1.36-fold) levels.

### 3.6. Cardiac Hydroxyproline Content

As demonstrated in Table 4, the CTR and DPPE groups displayed a non-significant change in the hydroxyproline concentrations. Meanwhile, the DOX group exhibited a substantial rise of about 2-fold when equated with the CTR quantity. In contrast, the DPPE + DOX group displayed a significant reduction (≈0.83-fold) compared to DOX values and a considerable enhancement (≈1.6-fold) compared to the CTR value.

### 3.7. Histopathological Results and Lesions Scoring

Figure 2 demonstrated the histomorphological results of H&E-stained cardiac tissue sections. Table 5 also illustrated the prevalence and severity of the identified pathological lesions in various treatments.

Heart tissues from the CTR (Figure 2a) and DPPE (Figure 2b) groups revealed normal histoarchitecture with well-organized and branched cardiac myofibers. The cardiomyocytes were closely arranged with oval centrally located nuclei, eosinophilic cytoplasm, and cross striations. In addition, minimal interstitial connective tissue and few fibroblasts were noticed. Meanwhile, the DOX group exhibited moderate to severe histological alterations and high lesion scores (Table 5), wherein disoriented cardiac myofibers with wavy appearance were evident. Furthermore, myocardial degenerative changes such as sarcoplasmic vacuolization (Figure 2c), myofibrillar flocculation, and fragmentation were noticed. Many cardiomyocytes showed Zenker’s degeneration. Meanwhile, others exhibited Zenker’s necrosis. Myocytolysis (Figure 2d) and multifocal zones of myocardial necrosis combined with infiltrations of mononuclear inflammatory cells (Figure 2e) were obvious. Additionally, hyperemia of interfibrillar blood vessels (Figure 1f), perivascular edema with inflammatory cell infiltrations (Figure 2g), intramyocardial edema, fibrin deposition, and focal areas of hemorrhage were noticed. There were interfibrillar infiltrations of active fibroblasts with a hypertrophic nucleus and marked myocardial and perivascular fibrosis. On the contrary, the DPPE + DOX group displayed a marked enhancement in cardiac tissue structure and integrity. Nevertheless, they were not identical to the CTR limits. Compared with the DOX group, the previously noted lesions were less in severities and distribution in the DPPE + DOX group (Figure 2h, Table 5).

### 3.8. Masson’s Trichrome Staining and Histomorphometric Findings

As illustrated in Figure 3, the cardiac tissue sections of the CTR (Figure 3a,e) and DPPE (Figure 3b,f) groups displayed normal spreading of greenish delicate collagen fibers in between the cardiomyocyte fibers and around the intramyocardial coronary vessels. They also revealed non-significant changes (*p* > 0.05) in the mean CVF% (Figure 3i) and PVCA% (Figure 3j). Conversely, the DOX group (Figure 3c,g) exhibited an apparent increase in the amount of collagen fiber deposition as well the mean CVF % (≈8.5-Fold) and PVCA% (≈3.09-fold) as compared to the CTR values (Figure 3i,j, respectively). However, the DPPE + DOX group revealed a marked reduction in collagen fiber deposition (Figure 3d,h). Meanwhile, the mean CVF % and PVCA% showed significant (*p* > 0.05) decrease with approximately 0.61-fold and 0.68-fold, respectively, when linked to DOX group values and significant (*p* > 0.05) increases with approximately 5.24 and 2.11-fold, correspondingly when equated to the CTR values (Figure 3i,j, respectively).

### 3.9. Immunohistochemical Analysis

For the expression of TGF-β1 in the cardiac tissue of the control groups, the CTR (Figure 4a) and DPPE (Figure 4b) groups revealed a normal expression of TGF-β1 (brown color). Both groups showed no significant alterations in the mean immune-stained area % of TGF-β1 (Figure 4e). Conversely, DOX-treated rats exhibited a noticeable increase in TGF-β1 expression (Figure 4c) with a substantial (*p* < 0.05) rise of the mean immune-stained area% (≈2.6-fold), as associated with the CTR group value (Figure 4e). However, DPPE + DOX-treated rats exhibited a conspicuous decrease in TGF-β1 expression (Figure 4d) with a substantial (*p* > 0.05) decline in mean immune-stained area% (≈0.7-fold) as compared to the DOX group value and a significant (*p* > 0.05) increase (≈1.84-fold) concerning CTR group quantity (Figure 4e).

Referring to cleaved caspase-3, Bax, and Bcl-2 expressions, the control (Figure 5a, Figure 6a and Figure 7a) and DPPE (Figure 5b, Figure 6b and Figure 7b respectively) groups revealed weak representation of cleaved caspase-3 and Bax. In addition, both groups showed a robust expression of Bcl-2 with dispersed, intensely brown stained immune-reactive cardiomyocytes. There were no significant alterations in the mean area% of cleaved caspase-3, Bax, and Bcl-2 immuno-stained cells (Figure 5e, Figure 6e and Figure 7e, respectively). Meanwhile, DOX-treated rats’ cardiac tissues displayed moderate to strong expression and immune-staining of cleaved caspase-3 (Figure 5c) and Bax (Figure 6c) and weak expression and immune-staining for Bcl-2 (Figure 7c). Relative to the CTR group values, the mean area% of immune-stained cells showed a substantial (*p* > 0.05) rise for cleaved caspase-3 (≈8.95-fold) and BAX (≈2.37-fold) and a significant decrease for Bcl-2 (≈0.3-fold) (Figure 5e, Figure 6e and Figure 7e, respectively).

The cardiac tissues of DPPE + DOX-treated rats exhibited weak expression and immune-staining of cleaved caspase-3 (Figure 5d) and Bax (Figure 6d). It also showed moderate to strong expression and immune-staining of Bcl-2 (Figure 7d). Constantly, the mean immune-stained areas % in DPPE + DOX group showed a significant (*p* < 0.05) reduction for cleaved caspase-3 (≈0.39-fold) and Bax (≈0.72-fold) and a considerable increase for Bcl-2 (≈1.9-fold) as associated with the DOX treated group values (Figure 5e, Figure 6e and Figure 7e, respectively). Meanwhile, they showed a substantial (*p* < 0.05) surge for cleaved caspase-3 (≈3.55-fold) and Bax (≈1.72-fold) and a significant decrease for Bcl-2 (≈0.59-fold), as equated with the CTR group (Figure 5e, Figure 6e and Figure 7e, respectively).

## 4. Discussion

Anthracyclines, including doxorubicin, play a crucial role in chemotherapy for the medication of numerous solid organ tumors and hematologic malignancy [12,16]. However, dose-based cardiotoxicity of anthracyclines is frequently reported to limit their therapeutic efficacy [5,65,66]. Since the DOX cardiotoxic effects are generally irreversible, searching for new protective approaches that could interrupt DOX-induced pathogenic events and confer protection against its cardiotoxicity should be developed [67]. Currently, the handout study is nearly the first to verify the meliorative potential of DPPE on DOX-induced cardiotoxicity. DOX cardiotoxicity is the ultimate obstacle to be solved to enhance its clinical usage [68]. In this work, rats were treated with 15 mg DOX/kg bw as a cumulative dose to mimic its chronic cardiotoxicity, as seen in clinical therapies [69]. Several pathways participate in DOX-induced cardiotoxicity.

However, the main mechanism involved is ROS generation, which causes peroxidation of lipids and depletion of antioxidant enzymes [35]. The cardiac tissue has many mitochondria because it needs much energy, making it more susceptible to DOX toxicity [67]. They have a high DOX affinity because their inner membrane encompasses cardiolipin, an anionic phospholipid with a high binding affinity to cationic DOX [67,70]. In mitochondria, cytochrome p450 reductase, xanthine oxidase, and NADPH dehydrogenase have to convert DOX into a semiquinone radical that interacts with molecular oxygen forming superoxide anion and an extra ROS [67]. Since free radicals act as the main contributor to DOX-induced cardiotoxicity, antioxidant compounds are known to be possible protective and therapeutic agents [71,72]. Dexrazoxane is the only synthetic medication used for cardiotoxicity prevention in clinical conditions [73,74]. DPP contains polyphenols and flavonoids with effective antioxidant and anti-inflammatory potentials that clarify its prospective use in many diseases. It also has antimicrobial, anti-coccidial, anti-apoptotic, and hepatoprotective potential [43,44,46]. Similarly, DPP is used as an anti-toxicant [42] and provides a cardio-preventive ability against isoproterenol-triggered myocardial infarction [42]. Its effect against DOX-induced cardiotoxicity has never been investigated.

The DOX group showed a scruffy appearance and exhibited 30% mortality. This result was parallel to those reported by Wu, et al. [75]. However, in DPPE + DOX-treated animals, there was relatively low mortality, reflecting that pre-cotreatment with DPPE might improve the survival of DOX-intoxicated rats. Additionally, the DOX persuaded a significant reduction in body weight [76,77], which might be due to decreased appetite, reduced protein synthesis, mucositis, and/or inadequate assimilation of nutrients [78,79,80]. The improved body weight in DPPE + DOX-treated animals mirrored the protective effects of DPPE. Cardiomyopathy caused by DOX is a shift from myocardial hypertrophy to heart failure [81]. In animals, myocardial hypertrophy is mainly assessed by measuring heart index weight [82]. In the current work, the DOX caused an increment in heart and relative heart weights and cardiomyocyte cross-sectional area, which indicated ventricular hypertrophy [75,81,83,84,85]. However, pre-cotreatment with DPPE revealed a reduction in the previous parameters, suggesting the ability of DPPE to maintain the normal integrity of cardiomyocytes. The cardiac enzymes ALT, AST, CK, CK-MB, LDH, CK, LDH, cTnI, and cTnT are a dynamic bioindicator of myocardial injury [35]. The DOX-induced cardiac damage was evident through the substantial rise of cardiac injury biomarkers: ALT, AST, LDH (not very specific biomarkers), CK, CK-MB, cTnI, and cTnT (more specific and sensitive biomarkers) activities, reflecting cardiomyocyte membrane disruption and extensive cardiomyocyte damage [59,76,86]. However, they were reduced following pre-cotreatment with DPPE, suggesting its competency to maintain the normal integrity of cardiac muscle and to inhibit DOX-induced myocardial damage [42]. An earlier study showed that antioxidant compounds could decrease cardiac function biomarkers in DOX-intoxicated rats [35].

The N-terminal pro-brain natriuretic peptide is a peptide produced to control blood pressure fluid equilibrium. It is liberated from the heart following ventricle volume expansion and/or pressure overload [87]. A large amount of NT-pro BNP is released into the blood during cardiac insufficiency, so it is considered a sensitive biomarker of congestive heart failure [88]. It is also a valuable predictor in patients with anthracycline chemotherapy as a vital biomarker of left ventricular dysfunction [89]. Herein, the DOX-induced, a dramatic increment of serum NT-pro BNP level, demonstrating that it can cause acute cardiac failure [35,90]. However, pre-cotreatment with DPPE lowered NT-pro BNP serum level, suggesting that DPPE may defend the heart from DOX-induced toxicity and cardiac damage.

The generation of large quantities of ROS and $O_{2}^{-}\;$ in DOX metabolism results in DNA and mitochondrial injury, therefore enhancing lipid peroxidation in the cell membrane and increased MDA levels in cardiac cells [91]. In turn, free radicals released in response to DOX can interfere with the balance between oxidative and antioxidants agents, followed by depletion of the endogenous myocardial antioxidant compounds (GSH) and enzymes (GPx, SOD, and CAT) [77]. Herein, DOX-intoxicated rats exhibited increased oxidative and nitrosative stresses, indicated by an increment of MDA and NO, and a substantial decline in antioxidant enzyme activity [76,92]. The recorded decline in the antioxidant enzyme activity could be attributed to their utilization in the fight against oxidative stress [35]. Remarkably, pre-cotreatment with DPPE diminished MDA and NO levels, and improved antioxidant activity in cardiac tissue, suggesting the antioxidant ability of DPPE against DOX-induced oxidative/nitrosative stress. Furthermore, DPPE expressed antioxidant activity and defensive mechanisms by restoring oxidative stress/antioxidant balance in several toxic modules [40,49,93]. Many earlier studies of phytochemical or antioxidant elements have demonstrated their ability to reduce lipid peroxidation and improve the value of antioxidant markers in DOX cardiotoxicity [21,35,71,76,94,95]. The mechanism for protecting the DPPE may include de-activating potentially toxic metabolites and free radicals and potentiation of antioxidant paths [96]. Other investigators claimed that DPPE incorporates considerable amounts of flavonoids, phytosterols, and carotenoids [42,97], which are antioxidants with redox activities acting as reducing agents ROS/NOS quenchers [98].

It is well established that increased free radicals output with excitotoxicity and lipid peroxidation accelerates inflammatory conciliators’ synthesis and thus activates the inflammatory response in the cardiac tissue [99]. NF-κB, a transcription factor, is involved in cell survival, inflammation, and immune responses. NF-κB p65 modulates the inflammatory responses, whereas its translocation to the nucleus enhances transcription of the pro-inflammatory cytokines, i.e., TNF-α, IL-1β, and IL-6 [100,101]. In turn, they provoke leukocyte infiltration into the myocardium and aggravate inflammatory injury [102]. Consistently, DOX stimulated NF-κB p65, TNF-α, and IL-1β production in cardiac tissues, which can also cause cardiomyocyte apoptosis by increasing ${\mathrm{Ca}}^{2+}$ store of heart muscle cells [26]. As well, DOX provoked a substantial rise in NF-κB level and correspondingly induced an increment in TNF-α and IL-1β quantities, reflecting enhanced inflammatory responses [71]. However, DPPE pre-cotreatment generated a substantial decline in NF-kB’s cardiac contents and the related downstream pro-inflammatory cytokines, TNF-α and IL-1β, indicating its potential to suppress the initiated inflammatory cascade.

Disruption of ${\mathrm{Ca}}^{2+}$ homeostasis is another pathway implicated in DOX-generated cardiac toxicity. It is documented that DOX cardiotoxicity is accompanied by a high overload of ${\mathrm{Ca}}^{2+}$ in cardiomyocytes, resulting in inadequate contraction and interference with ${\mathrm{Ca}}^{2+}$ regulation, thus triggering ROS generation and leading to cell dysfunction [103,104,105]. In the present work, DPPE was found to mitigate oxidative stress-mediated ${\mathrm{Ca}}^{2+}$ overload in DOX-challenged cardiac tissues. This may be attributed to DPPE’s antioxidant properties, which suppress ROS generation and consequently reduce such high ${\mathrm{Ca}}^{2+}$. Moreover, ROS and $O_{2}^{-}\;$ exaggerated cardiac dysfunction and mitochondrial damage induced in DOX therapy [17]. The formation of the Fe-anthracycline complex catalyzes the transformation of $H_{2}O_{2}$ to ${\mathrm{OH}}^{•}$ radicals, resulting in severe cytoskeleton injury and plasma membrane disruption followed by myofibril loss, sarcoplasmic reticulum dilation, and myocardial necrosis [24,106].

The findings of the cardiac tissues histopathological analysis supported the biochemical interpretations. Due to enlargement of the sarcoplasmic reticulum, DOX induced several degenerative changes in heart tissue, including myocardial hyalinization, and sarcoplasmic vacuolization [107], wavy myocardial fibers flocculation, and fragmentation. In addition, multifocal areas of myocardial necrosis, myofibrillar loss, inflammatory cell infiltrations, myocardial fibrosis, hemorrhage, vascular congestion, and interfibrillar edema were observed [71,74,100,108,109]. On the contrary, cardiac tissues of DPPE plus DOX-treated animals showed a marked improvement in cardiac tissue structure and integrity. According to the histomorphometric analysis, the previously declared lesions were less in incidence and severity. These remarks were comparable with other experiments that verified the potential of DPPE as anti-myocardial damage [42]. The literature reviewed that DPPE contains bioactive substances, including estradiol [110], stigmasterol [111], β-sitosterol [112], carotenoids lutein [113,114], δ-tocotrienol [115], and isorhamnetin [116], which have potential cardioprotective activities.

Fibrosis is a reparative reaction to DOX-induced cardiotoxicity [85]. The necrotic or apoptotic cardiomyocytes are replaced by overproduced collagen by fibroblast. However, it contributes to heart rigidity and instability [85,117]. DOX-induced cardiac fibrosis is based on the inflammatory and growth factors signaling paths regulated by TGF-β1. Increased oxidative stress and the subsequent antioxidant depletion and lipid peroxidation trigger tissue inflammation and necrosis and enhance tissue fibrogenesis progression [35,59,118]. TGF-β1 is another key factor in the regulation of collagen production in DOX-induced cardiomyopathy. The pro-fibrogenic cytokine, TGF-β1, is a proliferation-mediated fibrotic protein produced by cardiac myofibroblast and is responsible for cardiomyocyte hypertrophy, apoptosis, and fibrosis. It may stimulate cardiac fibroblastic hyperplasia, increased production of type I and III collagen fiber and fibronectin, and cause increased extracellular matrix and decreased extracellular matrix degradation through inhibition of collagen enzyme release [28,119]. In the current work, DOX elevated the cardiac hydroxyproline level, the major component of fibrillar collagen [59]. Additionally, DOX-induced remarkable hypertrophy of fibroblast increased collagen deposition, and fibrosis was further confirmed by Masson’s trichrome staining of the cardiac tissues [120]. DOX also increased the TGF-β1 expression in cardiac tissues [85,121,122,123]. However, DPPE pre-cotreatment induced a marked reduction in collagen fiber deposition between cardiac muscle fibers and around the intramyocardial coronary vessels, hydroxyproline content, and TGF-β1 expression in cardiac tissues. Therefore, DPPE has a potential aptitude to maintain the normal integrity of cardiac muscle and inhibit DOX-induced myocardial damage, which attenuated fibrosis development via modulation of fibrogenic genes.

The major regulators of apoptosis are Bcl-2 family members, which involve pro-apoptotic protein (Bax, caspase-3) and anti-apoptotic (Bcl-2) proteins [74]. During apoptosis, the Bcl-2 expression declines, while Bax and caspase-3 expressions rise [124]. Bax stimulation ensures cell damage by forming a pore in the mitochondrial membrane, leading to poly (ADP-ribose) polymerase cleavage and mitochondrial cytochrome-c induction, which mediate apoptosis [125]. Meanwhile, the Bcl-2 inhibits apoptosis by inhibiting mitochondrial permeability transition [126] in cardiomyocytes protecting mitochondrial structure and function [124]. Caspases are essential parts of the apoptotic process. Opened mitochondrial pores lead to mitochondrial cytochrome-C release, and activated caspase-3 triggers proteolytic degradation of cellular components death [127]. Herein, the immunohistochemical staining of myocardial tissues showed that DOX caused an increase in the cleaved caspase-3 and Bax expressions and decreased Bcl-2 expression, which reflected apoptosis’s role in DOX-induced cardiomyopathy [76,86,128]. Nevertheless, pre-cotreatment with DPPE reduced cleaved caspase-3 and Bax and enhanced Bcl-2 expressions, implying that DPPE’s anti-apoptotic activity could conserve myocardial integrity and mitigate myocardial damage.

## 5. Conclusions

Collectively, for the first time from these observations, it is indicated that date palm pollen ethanolic extract displayed an effective cardioprotective potential against doxorubicin-induced cardiac myopathy. The antifibrotic and anti-apoptotic mechanisms of DPPE were attributed to suppressing cardiac oxidative/nitrosative damage, pro-inflammatory cytokines production, and fibrogenic and apoptotic gene expressions, thereby reducing myocardial myopathy and detrimental structural alterations. Accordingly, DPPE is highly recommended as an adjunct to avert the toxic side-effects caused by doxorubicin.

## Figures and Tables

**Figure 1 animals-11-00886-f001:**
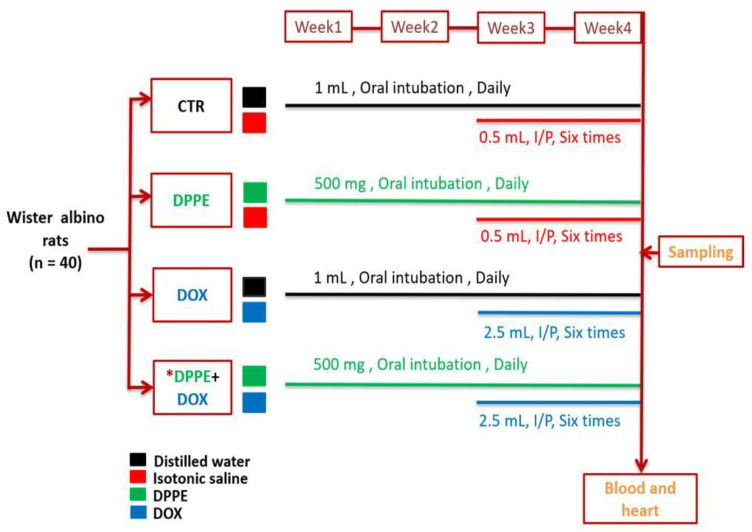
A schematic overview of the experimental protocol. CTR, Control group; DPPE, date palm pollen ethanolic extract-treated group; DOX, doxorubicin-treated group, DPPE + DOX, date palm pollen ethanolic extract- and doxorubicin treated group. * DPPE was given an hour before DOX administration.

**Figure 2 animals-11-00886-f002:**
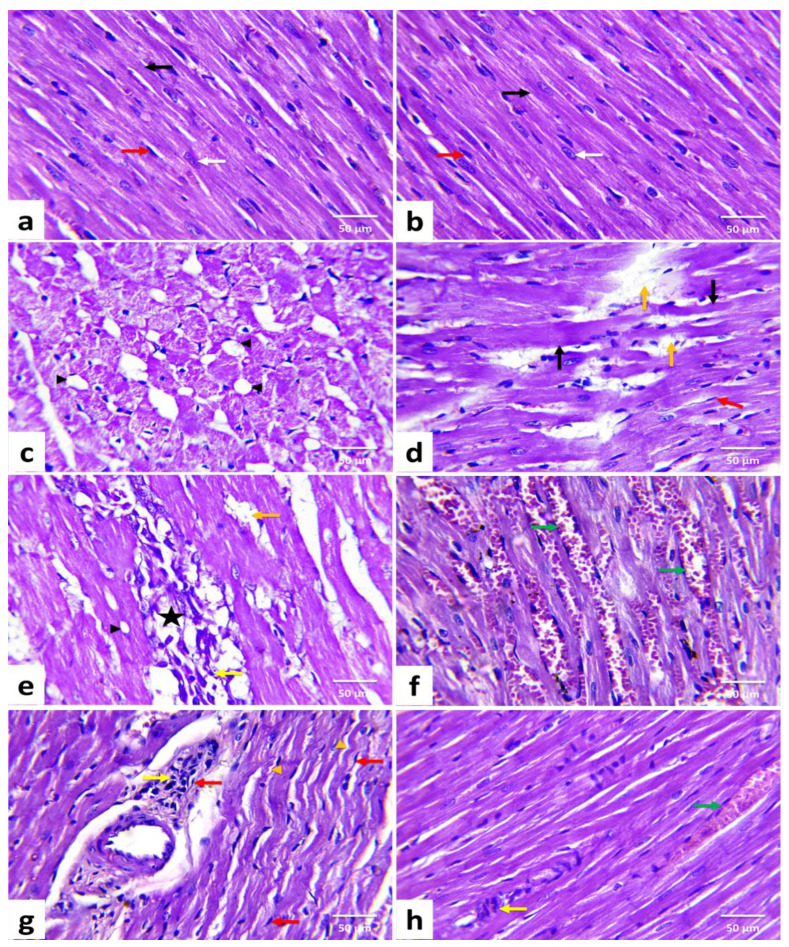
Histopathological changes of rats’ cardiac tissues (H&E, ×400). A rat from the control (**a**) and a rat from date palm pollen ethanolic extract-treated (**b**) groups, respectively showing normal histoarchitecture of the cardiomyocytes with well-organized and branched cardiac myofibers (black arrow), centrally located oval nuclei (white arrow), and minimal interstitial connective tissue with few interstitial fibroblasts (red arrow) in between. Doxorubicin-treated rats (**c**–**g**) showing vacuolization of the sarcoplasm (black arrowhead), Zenker’s necrosis (black arrow), wavy muscle fibers (orange arrowhead), loss of myofibrils (orange arrow), myocardial necrosis (star), mononuclear inflammatory cells infiltrations (yellow arrow), fibroblasts proliferation (red arrow), and hyperemic interstitial blood vessels (green arrow). DPPE + DOX-treated rat (**h**) showing marked improvement in muscle fibers striation, which almost looks like the control. However, minute areas of myocardial necrosis with inflammatory cells infiltration (yellow arrow) and hyperemic interstitial blood vessels (green arrow) are still evident. (*n* = 7). Each value is the average of seven observations.

**Figure 3 animals-11-00886-f003:**
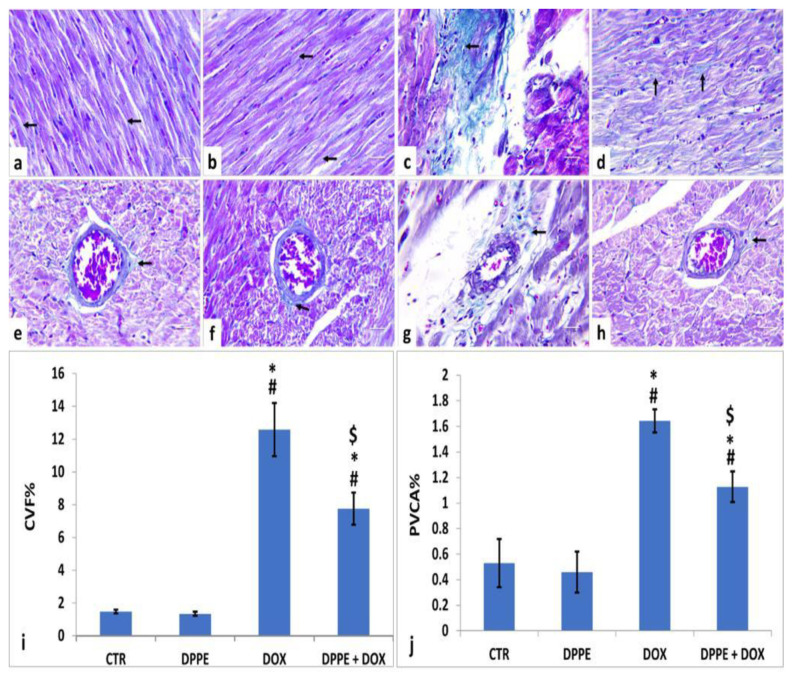
Histopathological changes of rats’ cardiac tissues (Masson’s trichrome, ×400, arrows: green stained collagen fibers). Rats from the control (**a**,**e**) and DPPE (**b**,**f**) groups, respectively showing scanty collagen fibers deposition. DOX treated rat (**c**,**g**) showing increased collagen deposition. DPPE + DOX-treated rat (**d**,**h**) showed a relative reduction in collagen deposition. Quantification of collagen volume fraction (CVF%), and perivascular collagen area (PVCA%) (**i**,**j**), respectively, ×400 across 10 different fields/section), *n* = 7 rat/group. Mean values were statistically differ from CTR (^#^
*p* < 0.05), DPPE (* *p* < 0.05), DOX (^$^
*p* < 0.05) groups.

**Figure 4 animals-11-00886-f004:**
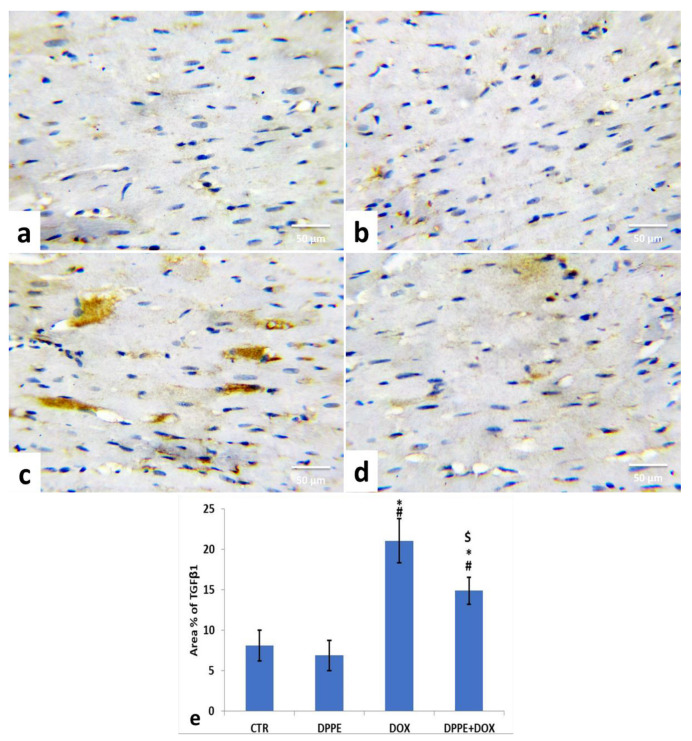
Immunohistochemical staining of transforming growth factor β1 (TGF-β1) in the experimental rats’ cardiac cells (IHC, ×400). A control (**a**), DPPE-treated (**b**) DOX-treated (**c**) and DPP + DOX-treated (**d**) rats. (**e**) Quantification of TGF-β1 expression, the immunohistochemical staining of TGF-β1 was measured as area percent (%) across 10 different fields/section, *n* = 7 rat/group. Mean values were statistically differed from CTR (^#^
*p* < 0.05), DPPE (* *p* < 0.05), DOX (^$^
*p* < 0.05) group.

**Figure 5 animals-11-00886-f005:**
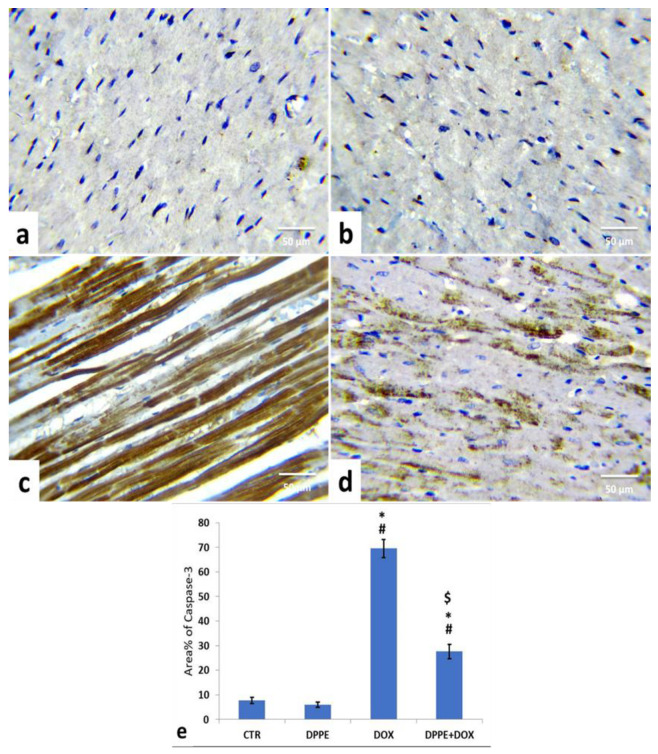
Immunohistochemical staining of cysteine aspartate specific protease-3 (cleaved caspase-3) in the cardiac cells of the experimental rats (IHC, ×400). A control (**a**), DPPE-treated (**b**) DOX-treated (**c**) and DPP + DOX-treated (**d**) rats. (**e**) Quantification of caspase-3 expression, the immunohistochemical staining of cleaved caspase-3 was measured as area percent (%) across 10 different fields/section, *n* = 7 rat/group. Mean values were statistically different from the CTR (^#^
*p* < 0.05), DPPE (* *p* < 0.05), and DOX (^$^
*p* < 0.05) groups.

**Figure 6 animals-11-00886-f006:**
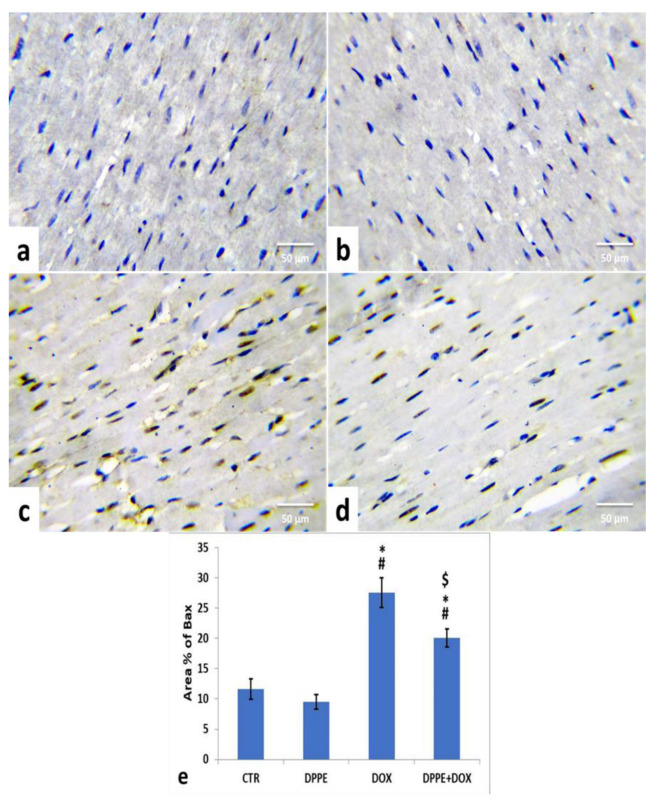
Immunohistochemical staining of Bcl2 associated X protein (Bax) in the experimental rats’ cardiac cells (IHC, ×400). A control (**a**), DPPE-treated (**b**) DOX-treated (**c**) and DPP + DOX-treated (**d**) rats. (**e**) Quantification of Bax expression, the immunohistochemical staining of Bax was measured as area percent (%) across 10 different fields/section, *n* = 7 rat/group. Mean values were statistically different from the CTR (^#^
*p* < 0.05), DPPE (* *p* < 0.05), and DOX (^$^
*p* < 0.05) groups.

**Figure 7 animals-11-00886-f007:**
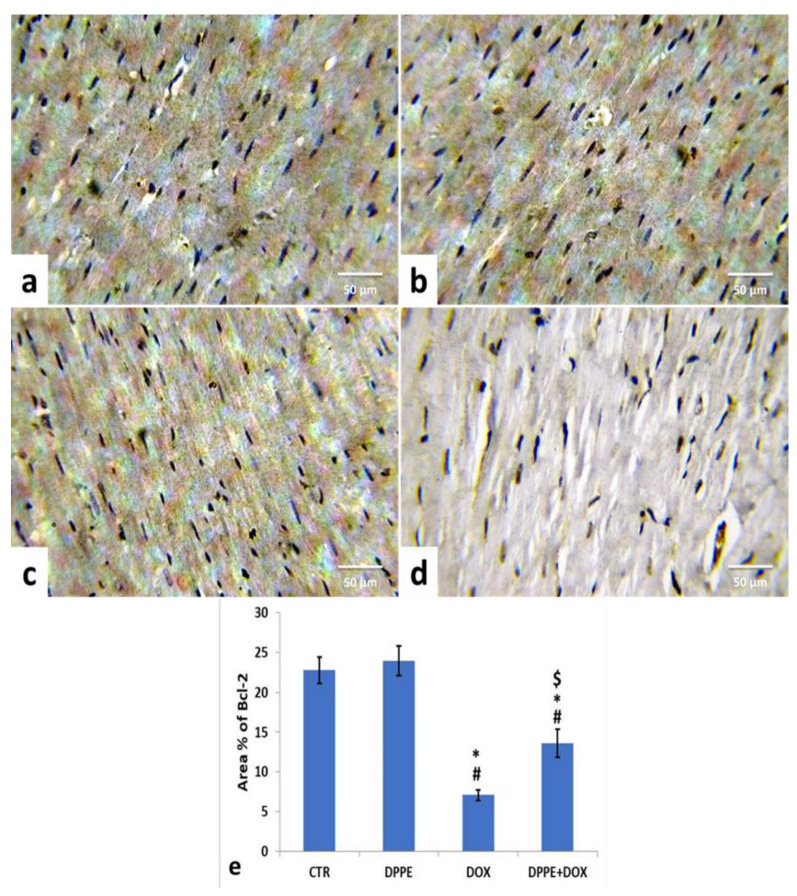
Immunohistochemical staining of B-cell lymphoma-2 (Bcl-2) in cardiac cells of the experimental rats (IHC, ×400). A control (**a**), DPPE-treated (**b**) DOX-treated (**c**) and DPP + DOX-treated (**d**) rats. (**e**) Quantification of Bcl-2 expression, the immunohistochemical staining of Bcl-2 was measured as area percent (%) across 10 different fields/section, *n* = 7 rat/group. Mean values were statistically different from the CTR (^#^
*p* < 0.05), DPPE (* *p* < 0.05), and DOX (^$^
*p* < 0.05) groups.

**Table 1 animals-11-00886-t001:** Effect of date palm pollen ethanolic extract (500 mg/kg bw/day) and/or doxorubicin (2.5 mg/kg bw/day) on body, heart, and relative heart weights myocyte cross-sectional area of control and treated rats.

Groups	Initial Body Weight (g)	Body Weight (g)	Heart Weight (g)	RHW (%)	Cardiomyocyte Cross-Sectional Area (μm^2^)
CTR	194 ± 4.5	218.43 ± 4.86	0.85 ± 0.013	0.52 ± 0.07	138.03 ± 3.90
DPPE	195 ± 5.2	221.85 ± 6.51	0.87 ± 0.023	0.49 ± 0.06	135.40 ± 2.65
DOX	190 ± 8.1	189.14 ± 8.36 *	1.36 ± 0.047 *	0.88 ± 0.11 *	210.87 ± 5.26 *
DOX + DPPE	192 ± 3.6	204.43 ± 2.72	1.04 ± 0.061 ^†^	0.63 ± 0.09	154.94 ± 2.06 ^†^

Control (CTR); date palm pollen ethanolic extract (DPPE); doxorubicin-treated (DOX); doxorubicin and date palm pollen ethanolic extract-treated (DOX + DPPE) groups, and relative heart weight (RHW). (*n* = 7). Each value is the average of 7 observations. Values are mean ± standard error (SEM). Mean values within the same columns were statistically different from CTR and DPPE (* *p* < 0.05). CTR, DPPE and DOX (^†^
*p* < 0.05).

**Table 2 animals-11-00886-t002:** Effect of date palm pollen ethanolic extract (500 mg/kg bw/day) and/or doxorubicin (2.5 mg/kg bw/day) on serum cardiac injury biomarkers and cardiac cytosolic calcium of control and treated rats.

Parameters	CTR	DPPE	DOX	DPPE + DOX
ALT (U/L)	69.26 ± 1.98	66.39 ± 4.25	105.09 ± 3.71 *	83.71 ± 3.04 ^†^
AST (U/L)	143.99 ± 4.74	140.25 ± 5.32	203.97 ± 6.98 *	175.36 ± 6.27 ^†^
LDH (U/L)	496.99 ± 18.28	485.76 ± 31.16	1237.62 ± 65.72 *	908.19 ± 9.61 ^†^
CK (U/L)	218.63 ± 17.16	203.71 ± 5.81	612.92 ± 7.97 *	431.50 ± 17.14 ^†^
CKMP (U/L)	458.99 ± 24.39	440.86 ± 32.52	934.56 ± 41.55 *	682.14 ± 37.44 ^†^
CTnI (pg/mL)	0.37 ± 0.05	0.30 ± 0.05	1.85 ± 0.38 *	1.07 ± 0.13 ^†^
CTnT (pg/mL)	0.91 ± 0.018	0.87 ± 0.03	3.15 ± 0.17 *	1.69 ± 0.15 ^†^
NT- ProBNP (pg/mL)	63.45 ± 3.73	57.31 ± 2.089	151.17 ± 3.44 *	96.55 ± 4.02 ^†^
Ca^+2^ (μg/g tissue)	32.80 ± 3.60	31.81 ± 2.95	77.35 ± 5.54 *	54.03 ± 5.17 ^†^

Control (CTR); date palm pollen ethanolic extract (DPPE); doxorubicin-treated (DOX); doxorubicin and date palm pollen ethanolic extract-treated (DOX + DPPE) groups; alanine aminotransferase (ALT); aspartate aminotransferase (AST); lactate dehydrogenase (LDH); creatine kinase (CK); creatine kinase-cardiac type isoenzyme (CK-MB); cardiac troponin I (CTnI); cardiac troponin T (CTnT); N-terminal pro-brain natriuretic peptide (NT-proBNP) and cardiac cytosolic calcium (Ca^+2^). (*n* = 7). Each value is the average of seven observations Values are mean ± standard error (SEM). Mean values within the same lines were statistically different from CTR and DPPE (* *p* < 0.05). CTR, DPPE and DOX (^†^
*p* < 0.05).

**Table 3 animals-11-00886-t003:** Effect of date palm pollen ethanolic extract (500 mg/kg bw/day) and/or doxorubicin (2.5 mg/kg bw/day) on cardiac nitro-oxidative stress lipid peroxidation in control and treated rats.

Parameters	CTR	DPPE	DOX	DPPE + DOX
NO (μmol/g tissue)	4.84 ± 0.93	3.32 ± 1.09	17.49 ± 2.45 *	10.25 ± 1.07 ^†^
MDA (nmol/g tissue)	37.35 ± 2.41	34.48 ± 2.73	69.52 ± 3.52 *	48.3726 ± 2.29 ^†^
GSH (mmol/g tissue)	25.27 ± 2.96	28.71 ± 1.65	11.73 ± 1.55 *	19.63 ± 0.85 ^†^
GPx (U/g tissue)	17.23 ± 1.48	19.08 ± 2.21	5.42 ± 1.26 *	11.84 ± 1.48 ^†^
CAT (U/g tissue)	34.61 ± 2.19	37.64 ± 2.38	18.92 ± 2.44 *	26.75 ± 2.70 ^†^
SOD (U/g tissue)	13.71 ± 1.318	15.45 ± 1.21	4.31 ± 1.22 *	9.85 ± 1.51 ^†^

Control (CTR); date palm pollen ethanolic extract (DPPE); doxorubicin-treated (DOX); doxorubicin date palm pollen ethanolic extract-treated (DOX + DPPE) groups; nitric oxide (NO); malondialdehyde (MDA); reduced glutathione (GSH); glutathione peroxidase (GPx); catalase (CAT) and superoxide dismutase (SOD). (*n* = 7). Each value is the average of 7 observations Values are mean ± standard error (SEM). Mean values within the same lines were statistically different from CTR and DPPE (* *p* < 0.05). CTR, DPPE and DOX (^†^
*p* < 0.05).

**Table 4 animals-11-00886-t004:** Effect of date palm pollen ethanolic extract (500 mg/kg bw/day) and/or doxorubicin (2.5 mg/kg bw/day) on cardiac inflammatory biomarkers hydroxyproline content in control and treated rats.

Parameters	CTR	DPPE	DOX	DPPE + DOX
NF-κB p65 (ng/g tissue)	90.23 ± 3.5	83.65 ± 3.8	238.09 ± 22.5 *	172.11 ± 14.8 ^†^
TNF-α (pg/g tissue)	34.29 ± 1.65	32.89 ± 1.89	80.75 ± 5.87 *	50.67 ± 4.81 ^†^
IL-1β (pg/g tissue)	79.81 ± 1.6	81.54 ± 1.7	138.85 ± 3.23 *	114.24 ± 3.65 ^†^
IL-6 (pg/g tissue)	47.20 ± 2.66	44.13 ± 1.71	87.63 ± 2.93 *	64.64 ± 2.21 ^†^
Hydroxyproline (μg/g tissue)	22.67 ± 2.03	20.61 ± 1.92	46.33 ± 2.21 *	38.3 ± 2.87 ^†^

Control (CTR); date palm pollen ethanolic extract (DPPE); doxorubicin-treated (DOX); doxorubicin date palm pollen ethanolic extract-treated (DOX + DPPE) groups; nuclear factor- kappa B (NF-κB p65); tumor necrosis factor-alpha (TNF-α), interleukin-1β (IL-1β) and interleukin-6 (IL-6). (*n* = 7). Each value is the average of seven observations Values are mean ± standard error (SEM). Mean values within the same lines were statistically different from CTR and DPPE (* *p* < 0.05). CTR, DPPE and DOX (^†^
*p* < 0.05).

**Table 5 animals-11-00886-t005:** Incidence and severity of cardiac histopathological lesions in the examined cardiac tissues in the control group and after various treatments in male Wistar albino rats.

Groups (*n* = 7)	Lesion Severity	Loss of Muscular Striations	Myocardial Vacuolation	Myocardial Necrosis	Myofibrillar Loss	Interstitial Inflammatory Cells Infiltrations	Hyperemic Blood Vessels	Interstitial Edema	Interfibrillar Hemorrhage	Myocardial Fibrosis
CTR	None	6	7	7	7	7	7	7	7	7
	(−)	−85.71%	−100%	−100%	−100%	−100%	−100%	−100%	−100%	−100%
	Mild	1	0	0	0	0	0	0	0	0
	(+)	−14.28%	0%	0%	0%	0%	0%	0%	0%	0%
	Moderate	0	0	0	0	0	0	0	0	0
	(++)	0%	0%	0%	0%	0%	0%	0%	0%	0%
	Severe	0	0	0	0	0	0	0	0	0
	(+++)	0%	0%	0%	0%	0%	0%	0%	0%	0%
DPPE	None	7	7	7	7	7	7	7	7	7
	(−)	−100%	−100%	−100%	−100%	−100%	−100%	−100%	−100%	−100%
	Mild	0	0	0	0	0	0	0	0	0
	(+)	0%	0%	0%	0%	0%	0%	0%	0%	0%
	Moderate	0	0	0	0	0	0	0	0	0
	(++)	0%	0%	0%	0%	0%	0%	0%	0%	0%
	Severe	0	0	0	0	0	0	0	0	0
	(+++)	0%	0%	0%	0%	0%	0%	0%	0%	0%
DOX	None	0	0	0	0	0	0	0	0	0
	(−)	0%	0%	0%	0%	0%	0%	0%	0%	0%
	Mild	0	0	0	1	0	0	1	1	1
	(+)	0%	0%	0%	−14.28%	0%	0%	−14.28%	−14.28%	−14.28%
	Moderate	2	6	5	6	7	5	6	5	6
	(++)	−28.57%	−85.71%	−71.42%	−85.71%	−100%	−71.42%	−85.71	−71.42%	−85.71%
	Severe	5	1	2	0	0	2	0	1	0
	(+++)	(71.42%)	−14.28%	−28.57%	0%	0%	−28.57%	0%	−14.28%	0%
DPPE + DOX	None	0	0	0	1	0	0	1	2	0
	(−)	0%	0%	0%	−14.28%	0%	0%	−14.28%	−28.57%	0%
	Mild	5	5	4	4	2	5	5	3	5
	(+)	(71.42%)	−71.42%	−57.14%	−57.14%	−28.57%	(71.42%)	−71.42%	−42.85%	−71.42%
	Moderate	2	2	3	2	5	2	1	2	2
	(++)	−28.57%	−28.57%	−42.85%	−28.57%	−71.42%	−28.57%	−14.28%	−28.57%	−28.57%
	Severe	0	0	0	0	0	0	0	0	0
	(+++)	0%	0%	0%	0%	0%	0%	0%	0%	0%

Control (CTR); date palm pollen ethanolic extract (DPPE); doxorubicin-treated (DOX); doxorubicin and date palm pollen ethanolic extract-treated (DOX + DPPE) groups. The severity of pathological lesions in different groups of rats was determined according to the percentage of tissue affected as: None (−): normal histology with zero involvement of the examined field, Mild (+):5–25% of the examined field is involved, Moderate (++): 26–50% of the examined field is involved Severe (+++): ≥50% of the examined field is involved. Incidence is the number of rats with lesions per total examined. (*n* = 7).

## Data Availability

All data sets obtained and evaluated during the current study are available upon appropriate request from the corresponding author.

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
