# Peer review of "Date Palm Pollen Extract Avert Doxorubicin-Induced Cardiomyopathy Fibrosis and Associated Oxidative/Nitrosative Stress, Inflammatory Cascade, and Apoptosis-Targeting Bax/Bcl-2 and Caspase-3 Signaling Pathways"

_animals, 2021, doi:10.3390/ani11030886_

Round 1

Reviewer 1 Report

This study was intended to explore the potential meliorative effect of date palm pollen ethanolic extract (DPPE) against DOX-induced cardiotoxicity and the mechanisms underlying it”.

It is an interesting work and whose results are promising for in another phase to know which molecules are responsible for the beneficial effects of DDP extracts as an adjunct to chemotherapeutics. As strengths I point out the manuscript as a whole. In fact, is very well written, its sections are presented in a balanced, coherent and with a very successful sequence of subjects. On the other hand, I would like to emphasize the remarkable amount of study approaches/markers. The results are very well presented and supported in figures and tables, including photomicrographs (some of immunohistochemistry are of lower quality). The data are properly and deeply discussed based on a considerable number of references. I recommend publishing the manuscript, but I would like to be clarified about some questions and make some suggestions.

Comment 1: Given the number of parameters/markers studied, of which the authors speak throughout the text, I suggest a list of abbreviations/acronyms at the beginning of the manuscript. I was registering, I'm not sure the list is complete:

ALT:

AST:

Bax:

Bcl2:

BW: body weight

CAT: catalase

CHF: congestive heart failure

CK:

CK-MB:

CVF: collagen volume fraction

cTnI: Serum cardiac troponin I

cTnT: Troponin T

DOX: Doxorubicin

DPPE: date palm pollen ethanolic extract

GPx: glutathione peroxidase

GSH:

IL-1β: interleukin-1 94 beta

IL-6: interleukin-6

LDH:

MDA:

NO:

NT-pro BNP: N-terminal pro-brain natriuetic peptide

PVCA: collagen area

RCF: radius centrifugation force

RNS: reactive nitrogen species

ROS: and oxidative stress induction

SOD: superoxide dismutase

TNF-α: tumor necrosis factor-alpha

UDP: up and down procedure

Comment 2: Authors should carefully read the manuscript and correct punctuation, uppercase/lowercase letters, link between periods/sentences, etc.

Line 17 (Simple Summary): Double extract in the sentence “The use of date palm pollen ethanolic extract (DPPE) extract…”

ABSTRACT

Comment 3 (line 33 and throughout the document): replace bwt by BW.

Comment 4 (Keywords): I suggest adding Date palm (pollen extract) and Doxorubicin.

1. INTRODUCTION

Line 71: “Gens that can regulate…” »» “Genes that can regulate…”

Line 118: Phoenix dactylifera L. in italic Phoenix dactylifera L.

Comment 5 (line 137): I suggest finishing the Introduction with a sentence with the objectives of the research. Maybe as in the Abstract:

This study was intended to explore the potential meliorative effect of date palm pollen ethanolic extract (DPPE) against DOX-induced cardiotoxicity and the mechanisms underlying it”.

2. MATERIALS AND METHODS

Comment 6 (lines 167-170): It seems to me that there is no link between these two phrases. “The obtained supernatant was evaporated at 40°C in a rotary evaporator under vacuum till full dryness. The final dry extract and stock solution preserved in dark glass bottles in the refrigerator at 4°C for further analysis”.

Comment 7 (Line 174): I don't understand “Thermo-acute tocixity”

Comment 8 (line 204): I suggest “…were randomly distributed…”

Comment 9 (line 204): The authors must also refer the mean BW in each group at the beginning of the experiment, and then include it in Table 1 to find out if the final values explain the effects of the treatments.

Comment 10 (line 208): please use the same time scale 4 weeks “…bwt daily for one month…”

Comment 11: “DOX injected intraperitoneally at a dose of 2.5 mg/kg bw”. Is this the established dose?

Comment 12: What was the administration time of the day? Always at the same hour of the day?

Comment 13 (lines 247-254): Please tell how many hearts went to one and the other technique.

Comment 14 (line 256): The heart weight has not been estimated, so it should be: “At the end of the experimentation, in each rat, the body and heart weight were recorded. The relative heart weights (RHW) were estimated using the following formula: …”

Line 327: “…sections were simmered…” incubated

Comment 15 (line 337):  What controls, negative and positive, were used, in IHC, for each of the antibodies?

Comment 16 (lines 352-352): In the PVCA (%) I understood that this parameter was used in another paper, but for me it might make more sense to consider the ratio (area occupied by the collagen/ total area of the vessel section) x 100.

Comment 17 (lines 354-356): “Using images of immunostained slides, the area percentage (%) of TGF-β1, cleaved caspase-3, Bax and Bcl-2 immunopositive cardiomyocytes were estimated [64]” How was this % calculated? Positive area (brown)/field area? I read after in the legend of Figure 4, but it must be explicit in the Materials and Methods (“…as area percent (%) across 10 different fields/section…)

3. RESULTS

Comment 18 (Line 364): Median lethal dose, mortality and survival rates” must be identified as a sub-chapter, so “3.1 Median lethal dose, mortality and survival rates”.

Comment 19 (Table 1): The comparison of final weights is somewhat compromised without knowing the initial weights. At the beginning of the experiment, did the average weights of the 4 groups not have significant differences? Thus, the initial BW weights must be presented in the Table 1 and this aspect must be referred to in the Materials and Methods. Moreover, the effects of the treatments are evident.

Comment 20: For all Tables, refer the N of rats and say that each value is the average of X observations.

Comment 21: The authors change the presentation logic between Tables, I think it should always be the same. Perhaps except for Table 5 (due to the volume of data), I think they would be better with the evaluated parameters presented in the first column on the left (such as Tables 2, 3 and 4) and in the first line the groups. Tables 1 and 5 are presented differently. However, if the authors decide to keep as are in manuscript they should correct the footnotes of Tables 2, 3 and 4 to “The mean differences between the values bearing different superscript letters within the same lines are statistically significant (p < 0.05)”. (The comparisons are among lines and not columns).

4. DISCUSSION

Comment 22: The doses used, and for just 4 weeks, in rats, can be extrapolated to humans in possible longer treatments? Shouldn't gastric side effects also be considered? Is this product already used in humans or are the results only for animal models?

Reviewer 2 Report

Dear Authors, 

There are no clarifications of the terms 'a','b''c' used in all the tables presented. Please change all statistical significance values to the internationally accepted "*" method. 

All table and figure legends must include the replicate number (n).

Many textual edits required in the manuscript. For e.g. Figure 1 should have "week 1" rather than "weak 1". Line 204: why accidental, is this supposed to mean "randomly"?

For the conclusion part: it is okay, but needs english restructuring, some sentences look unreasonable 

For the data and analysis method, statistical analyses need to be corrected to the international format. 

discussion is a bit too long, can be cut down 。
